# ADABOOST-BASED LOCAL-FOREST ADVERSARIAL LEARNING FOR IMBALANCED DOMAIN ADAPTATION

## ABSTRACT

Class imbalance poses a significant challenge in unsupervised domain adaptation (UDA). We propose Adaboost-based Local-Forest Adversarial Learning (AL-FAL), a framework that leverages sample-wise label matching rates to guide both adaptive sampling and interpolation-based generation. ALFADA first samples informative instances and constructs a local discriminative forest (LDF) via clustering to enable fine-grained regional alignment on the basis of the global discriminator in the framework of domain adversarial learning. To further enhance adaptation for minority classes, a Boosted Pairwise Interpolation Generator (BPIG) synthesizes interpolation samples between high-weight source and confident target instances. These auxiliary samples are optimized through an adversarial learning-based exploration mechanics to explore challenging regions. Experiments demonstrate that ALFADA consistently outperforms existing state-of-the-art methods on imbalanced domain adaptation benchmarks.

## 1 INTRODUCTION

Unsupervised Domain Adaptation (UDA) aims to transfer knowledge from a labeled source domain to an unlabeled target domain by mitigating the distributional discrepancy between them. While recent UDA techniques (Ganin & Lempitsky, 2016; Long et al., 2018) have achieved promising results under the assumption of balanced class distributions, this assumption is often violated in real-world scenarios. Both the source and target domains frequently exhibit significant class imbalance, where a few categories dominate the data while others remain under-represented. This exacerbates alignment difficulty and leads to degraded performance on minority classes, a problem referred to as Class-Imbalanced Domain Adaptation (CDA) (Tan et al., 2020; Wu et al., 2019b).

To address this, existing works have primarily followed two directions: pseudo-label-based self-training and data augmentation. Pseudo-labeling methods (Tan et al., 2020; Jiang et al., 2020; Prabhu et al., 2021) generate class predictions on unlabeled target samples and retrain the model in a self-supervised fashion. Some methods incorporate class-aware weighting or region-specific alignment based on the pseudo-label (Alcover-Couso et al., 2025; Liu et al., 2023b). However, these methods often rely on static or pre-defined alignment structures, limiting their ability to adaptively emphasize hard or minority-class regions during training.

To complement pseudo-labeling, a growing body of research has focused on data augmentation, especially for minority classes. Adversarial augmentation methods (Miyato et al., 2018; Shu et al., 2018) and sample interpolation approaches (Shi et al., 2022b) have shown that generating synthetic samples along decision boundaries can enrich target domain coverage and enhance classifier robustness. Pairwise Adversarial Training (PAT) (Shi et al., 2022b), for example, interpolates between a source and a target sample of the same class to generate informative intermediate samples. However, most existing augmentation methods do not explicitly identify alignment difficulty or underperforming regions, and often assume correct sample pairing which may not hold under pseudo-label noise and distributional shift.

To bridge these gaps, we propose a unified framework named Adaboost-based Local-Forest Adversarial Domain Adaptation (ALFADA), which jointly addresses class imbalance, pseudo-label noise, and fine-grained domain misalignment. ALFADA is composed of three tightly coupled components that work synergistically to enhance adaptation performance under class-imbalanced settings.

First, we introduce an adaptive sample reweighting mechanism inspired by AdaBoost. In each epoch of the training, we estimate the per-class matching error rates (MER) to reflect the alignment quality across domains. These rates are used to dynamically assign weights to individual samples, emphasizing hard-to-align or minority-class instances so as to push the model to allocate more attention to informative and underrepresented samples.

Building upon the weighted sampling, we then construct a local discriminative forest (LDF) on the basis of the framework of adversarial domain adaptation. Instead of relying solely on a single global domain discriminator, ALFADA partitions the feature space into multiple clusters and trains a separate local domain discriminator adaptively for each cluster alongside the global one. This forest-like structure captures fine-grained regional discrepancies between domains and preserves local semantic consistency, which is especially beneficial for the alignment of rare categories that may otherwise be overwhelmed by dominant classes in global alignment.

Finally, to further enhance the feature space coverage of minority classes, we propose a Boosted Pairwise Interpolation Generator (BPIG). BPIG synthesizes new samples by interpolating between high-weighted source instances and confident target samples of the same predicted class. These interpolated samples serve as bridges between sparse regions in the joint feature space. Furthermore, we design an adversarial learning-based exploration mechanics to optimize the generated samples in order to explore the challenging regions. BPIG ensures that the generated samples lie near class boundaries and guide the classifier to learn more discriminative decision surfaces, particularly for underrepresented or ambiguous classes.

In summary, our contributions are threefold:

- Introduce an adaptive reweighting mechanism that guides all stages of training to focus on minority and/or hard-to-align samples.

- Construct a local discriminative forest via adaptive sampling and clustering, leveraging ensemble effects to realize fine-grained feature alignment and prevent overfitting towards the majority classes.

- Propose a boosted interpolation generator that synthesizes informative source-target samples to densify low-density regions and enhance minority-class learning.

## 2 RELATED WORK

**Class-Imbalanced Domain Adaptation.** Class-imbalanced domain adaptation (CDA) poses additional challenges due to biased prediction toward frequent classes. Early works like COAL (Tan et al., 2020) and Implicit Class Conditioning (Jiang et al., 2020) leverage pseudo-labels to enforce class balance during self-training. ACE (Wu et al., 2019b) introduces asymmetrical alignment losses, while SENTRY (Prabhu et al., 2021) uses consistency regularization to filter noisy pseudo-labels. Recent studies such as WUDA (Liu et al., 2023a) refines weighting mechanisms or fusion strategies to better handle imbalance. GBW (Huang et al., 2023), dynamically weight classes by gradient magnitude. In parallel, advanced reweighting techniques have been proposed to tackle class imbalance. For example, Guo et al. (Guo et al., 2022) formulate sample weighting as an optimal transport problem to directly balance class distributions without expensive bilevel optimization. However, all these approaches heavily rely on pseudo-label quality, which is particularly unreliable for rare classes.

**Multi-discriminator adversarial learning** Adversarial-based UDA frameworks like DANN (Ganin & Lempitsky, 2016) focus on marginal alignment using a global domain discriminator. MADA (Pei et al., 2018) extends this idea by employing class-wise discriminators to capture conditional distributions. COT (Liu et al., 2023b) uses clustering and optimal transport to align cluster-level prototypes. To better address class-specific misalignment, some works introduce fine-grained alignment strategies. For example, CaCo (Huang et al., 2022) employs category-aware contrastive objectives to preserve class-wise feature separation during alignment, and an attention-based class-conditioned aligner (ACIA) (Belal et al., 2025) for multi-source DA was proposed to specifically improve alignment of minority classes in object detection. Such approaches highlight the benefit of local discriminators focused on challenging regions, though they are not yet widely integrated into standard UDA pipelines.

**Data Augmentation and Sample Generation.** Data augmentation plays a key role in enhancing target coverage, especially for rare categories. SMOTE-style oversampling has been widely applied in traditional long-tailed learning, but less so in UDA. Virtual Adversarial Training (VAT) (Miyato et al., 2018; Shu et al., 2018) introduces input perturbations to enforce smoothness. PAT (Shi et al., 2022b) advances this by generating synthetic samples through source–target interpolation. Recently, generative data augmentation has made progress by leveraging large pre-trained models to bridge domains. For instance, DoGE (Wang et al., 2024) uses a diffusion-based strategy to generate target-style synthetic samples by encoding the source–target domain gap, yielding performance gains without requiring explicit source-target pairs. However, these modern augmentation techniques either assume accurate source–target pairing or ignore region-specific alignment difficulty.

## 3 METHODOLOGY

### 3.1 PROBLEM SETUP AND FRAMEWORK OVERVIEW

We address the problem of *unsupervised domain adaptation (UDA)* under *class imbalance*, where both the source and target domains exhibit long-tailed label distributions. Formally, let the labeled source domain be denoted as $\mathcal{D}_s = \{(\boldsymbol{x}_i^s, y_i^s)\}_{i=1}^{N_s}$, and the unlabeled target domain as $\mathcal{D}_t = \{\boldsymbol{x}_j^t\}_{j=1}^{N_t}$, where $y_i^s \in \{1, \ldots, C\}$ and $C$ is the total number of categories. The goal is to learn a feature extractor $g(\cdot)$ and classifier $f(\cdot)$ such that $f(g(\boldsymbol{x}_j^t))$ yields accurate predictions for target samples.

The ALFADA framework consists of five key components: Feature Extractor ($g$): Maps both source and target samples into a shared feature space; Local discriminative forest: Constructs $K$ local domain discriminators via weighted K-means clustering, supplemented by a global discriminator; Classifier ($f$): Provides class predictions for both source supervision and target pseudo-labeling. An overview of the proposed framework is shown in Figure 1.

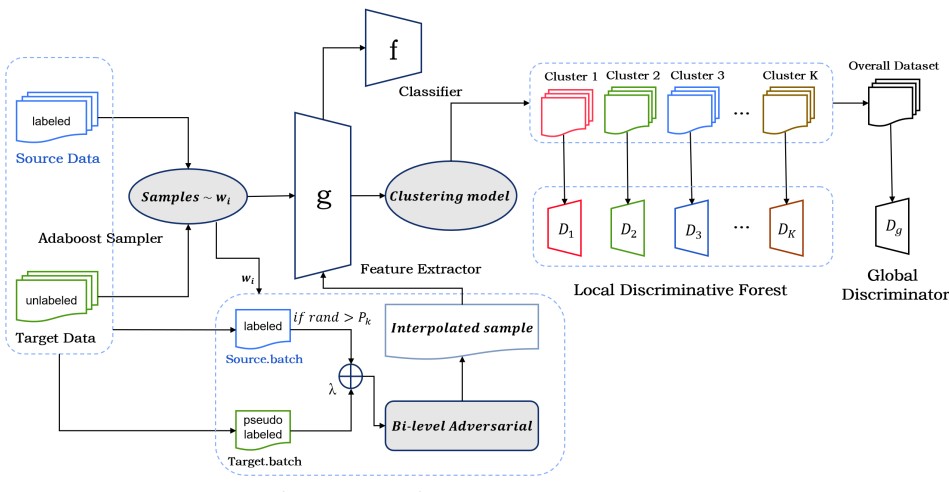

Figure 1: Overall architecture of the ALFADA framework.

### 3.2 ADAPTIVE MATCHING ERROR RATE SAMPLING

To effectively highlight hard-to-align and minority-class samples under class imbalance, we introduce an adaptive *matching error rate*–based weighting scheme. This mechanism serves as the foundation for weighted sampling and downstream clustered domain alignment.

### 3.2.1 PSEUDO-LABEL ASSIGNMENT THRESHOLD

First, we define the overall sample set $\mathcal{D} = \{(\boldsymbol{x}_i, y_i^{\text{match}}, d_i)\}_{i=1}^{N}$ where $\boldsymbol{x}_i$ is the input sample, $y_i^{\text{match}}$ is the corresponding matching label, and $d_i \in \{0, 1\}$ indicates the domain (0 for source, 1 for target). The definition of $y_i^{\text{match}}$ depends on the domain to which $\boldsymbol{x}_i$ belongs. For source domain samples, $y_i^{\text{match}}$ is simply the ground-truth label $y_i^s$. For target domain samples, we rely on pseudo-labels $\hat{y}_i^t$ predicted by the classifier trained in the previous epoch. To maintain a stable training, only target samples with pseudo-label confidence above a fixed threshold $\tau$ are included. Specifically, we set:

$$y_i^{\text{match}} = \begin{cases} y_i^s & \text{if } \boldsymbol{x}_i \in \mathcal{D}_s \\ \hat{y}_i^t & \text{if } \boldsymbol{x}_i \in \mathcal{D}_t \text{ and } p(\hat{y}_i^t) \geq \tau \end{cases}$$

Target samples with confidence below $\tau$ are excluded from matching rate computation and weight updates in order to rule out the noise in the pseudo-labels.

### 3.2.2 DEFINITION OF MATCHING ERROR RATE

We define the *matching error rate* (MER) $\varepsilon_c$ for each class $c \in \{1, \ldots, C\}$ as follows:

$$\varepsilon_c = \frac{\sum_{i:y_i^{\text{match}}=c} w_i \cdot \mathbb{I}(f(g(\boldsymbol{x}_i)) \neq y_i^{\text{match}})}{\sum_{i:y_i^{\text{match}}=c} w_i} \tag{1}$$

Here, $f(g(\boldsymbol{x}_i))$ denotes the predicted class for input $\boldsymbol{x}_i$, $w_i$ is the current sample weight, and $\mathbb{I}(\cdot)$ is the indicator function.

### 3.2.3 ADAPTIVE SAMPLING

We initialize all weights uniformly as $w_i^{(0)} = 1/N$ at the beginning and normalize after each update to ensure that $\sum_i w_i = 1$. Once the matching rate $\varepsilon_c$ is computed at $t$-th epoch, we derive a class-level coefficient $\alpha_c$ using an Adaboost-style formulation:

$$\alpha_c = \frac{1}{2} \ln \left( \frac{1 - \varepsilon_c}{\varepsilon_c} \right) \tag{2}$$

Based on $\alpha_c$, we update the adaptive weight for each class via:

$$w_i^{(t+1)} = w_i^{(t)} \cdot \exp \left( \alpha_{y_i^{\text{match}}} \cdot \mathbb{I}(f(g(\boldsymbol{x}_i)) \neq y_i^{\text{match}}) \right) \tag{3}$$

This update rule increases the weight $w_i$ for samples that are misclassified, especially those belonging to harder classes with larger $\alpha_c$. We initialize all weights uniformly as $w_i^{(0)} = 1/N$ and normalize after each update to ensure that $\sum_i w_i = 1$.

Then we use these adaptive weights to construct a rebalanced training dataset $\mathcal{D}_w$ through sampling with replacement. Each sample $\boldsymbol{x}_i$ is selected with probability $w_i$:

$$P(\boldsymbol{x}_i \in \mathcal{D}_w) = w_i$$

This mechanism implicitly amplifies the contribution of underrepresented and hard-to-classify samples. As a result, the constructed dataset $\mathcal{D}_w$ better reflects the class distribution difficulties and provides a more informative basis for downstream clustering and adversarial training.

## 3.3 LOCAL DISCRIMINATIVE FOREST

To capture fine-grained domain discrepancies under class imbalance, we introduce a Local Discriminative Forest (LDF) on the basis of DANN. The LDF consists of multiple local domain discriminators, each trained on a specific sub-region of the feature space. This design compensates for the limitations of a single global discriminator, which tends to align dominant modes of the distribution while overlooking minority-class features.

The domain discriminator forest consists of $K$ independent sub-domain discriminators $\{D_1, D_2, \ldots, D_K\}$ and a global discriminator $D_{\text{global}}$. Each sub-domain discriminator $D_i$ is trained on a distinct dataset $\mathcal{C}_i$, sampled by weighted K-means clustering.

### 3.3.1 MER-WEIGHTED CLUSTERING

To enable fine-grained domain alignment, we partition the weighted training set $\mathcal{D}_w$ into $K$ subdomains using a weighted K-means algorithm. Each sample $\boldsymbol{x}_i$ is represented by its feature $\boldsymbol{z}_i = g(\boldsymbol{x}_i)$ and associated adaptive weight $w_i$ from Section 2.2. By incorporating $w_i$ into the clustering process, samples from minority classes or hard-to-align regions exert greater influence on cluster formation.

We first initialize $K$ cluster centroids $\{\boldsymbol{\mu}_1^{(0)}, \ldots, \boldsymbol{\mu}_K^{(0)}\}$ by sampling features from $\mathcal{D}_w$ with probability proportional to $w_i$. This encourages initialization near informative, underrepresented regions.

Clustering proceeds by alternately assigning each $\boldsymbol{z}_i$ to the nearest centroid based on weighted distance

$$\text{dist}_w(\boldsymbol{z}_i, \boldsymbol{\mu}_k^{(p)}) = w_i \cdot \|\boldsymbol{z}_i - \boldsymbol{\mu}_k^{(p)}\|_2^2, \tag{4}$$

and updating each centroid as the weighted average of its assigned features:

$$\boldsymbol{\mu}_k^{(p+1)} = \frac{\sum_{\boldsymbol{z}_i \in \mathcal{C}_k} w_i \cdot \boldsymbol{z}_i}{\sum_{\boldsymbol{z}_i \in \mathcal{C}_k} w_i}. \tag{5}$$

The procedure terminates after a fixed number of iterations $T_{\text{cluster}}$ or when all centroid updates fall below a threshold $\epsilon$.

The resulting partition $\{\mathcal{C}_1, \ldots, \mathcal{C}_K\}$ defines $K$ local feature regions. Each cluster $\mathcal{C}_k$ is assigned a dedicated domain discriminator $D_k$, responsible for aligning source and target distributions within that region. To ensure that the local domain discriminators remain aligned with the current feature distribution and balance the precision-efficiency trade-off, we perform this weighted clustering every $T$ epochs.

### 3.3.2 ADAPTIVE DOMAIN ALIGNMENT

Once the updated clusters $\{\mathcal{C}_1, \ldots, \mathcal{C}_K\}$ are formed, we re-initialize the local discriminators $\{D_k\}_{k=1}^K$, each of which is trained to distinguish source from target samples within its assigned cluster. Since some subdomains may be more difficult to align than others (e.g., those dominated by minority classes or exhibiting more shift), we assign each $D_k$ an adaptive importance weight based on its domain classification performance.

Each local discriminator $D_k$ is trained to distinguish source from target samples within its corresponding cluster $\mathcal{C}_k$, using a binary cross-entropy loss defined as:

$$\mathcal{L}_{\text{local},k} = -\frac{1}{N_k} \sum_{i \in \mathcal{C}_k} \left[ d_i \log(D_k(\boldsymbol{z}_i)) + (1 - d_i) \log(1 - D_k(\boldsymbol{z}_i)) \right] \tag{6}$$

where $d_i \in \{0, 1\}$ denotes the domain label (0 for source, 1 for target), $N_k$ is the sample size of $\mathcal{C}_k$ and $D_k(\boldsymbol{z}_i)$ is the predicted domain probability output by the $k$-th local discriminator.

To avoid overfitting to localized patterns and ensure global distribution alignment, we keep the original global domain discriminator $D_{\text{global}}$, which is trained on a overall dataset $\mathcal{D}_w$ at this epoch. The resulted loss function is defined as:

$$\mathcal{L}_{\text{global}} = -\frac{1}{N} \sum_{i=1}^N \left[ d_i \log(\hat{d}_{i,\text{global}}) + (1 - d_i) \log(1 - \hat{d}_{i,\text{global}}) \right] \tag{7}$$

where $N = \sum_{k=1}^K N_k$ and $\hat{d}_{i,\text{global}} = D_{\text{global}}(\boldsymbol{z}_i)$ , which is the predicted domain label probability by $D_{\text{global}}$.

Specifically, in order to direct the model's focus toward more challenging areas, we evaluate each local discriminator $D_k$ on a validation subset $\mathcal{V}_k \subseteq \mathcal{C}_k$ and compute its domain error rate:

$$\delta_k = \frac{1}{|\mathcal{V}_k|} \sum_{i \in \mathcal{V}_k} \mathbb{I}(D_k(\boldsymbol{z}_i) \neq d_i), \tag{8}$$

where $d_i \in \{0, 1\}$ is the ground-truth domain label (source or target) of sample $\boldsymbol{x}_i$. The corresponding importance weight $\beta_k$ is then computed via:

$$\beta_k = \frac{1}{2} \ln \left( \frac{1 - \delta_k}{\delta_k} \right). \tag{9}$$

Intuitively, regions with high $\delta_k$ represent under-adapted subdomains, which should contribute more strongly to the overall alignment process.

The overall domain alignment loss integrates both local and global discriminators. The local losses are aggregated using the importance weights $\beta_k$ introduced in the previous section:

$$\mathcal{L}_{\text{domain}}(\{\theta_{D_i}\}, \theta_{D_{\text{global}}}) = \sum_{k=1}^{K} \beta_k \mathcal{L}_{\text{local},k} + \mathcal{L}_{\text{global}}. \tag{10}$$

Maximization of Eq. (16) allows each $D_k$ to focus on aligning its local feature region, while $D_{\text{global}}$ ensures that global distribution-level shift is corrected. The adaptive weighting $\beta_k$ encourages the model to emphasize hard-to-align regions, which often correspond to underrepresented or minority class subdomains.

### 3.4 BOOSTED PAIRWISE INTERPOLATION GENERATOR

To alleviate class imbalance and local domain mismatch, we propose a Boosted Pairwise Interpolation Generator (BPIG) that synthesizes auxiliary samples by interpolating between source-target sample pairs of the same class. These generated samples enhance feature continuity across domains and serve as informative training instances for adversarial alignment.

#### 3.4.1 GENERATION TRIGGER

We prioritize sample generation for minority and hard-to-align classes using an adaptive threshold probability $P_k$ for each class $k$:

$$P_k = \frac{n_k / n_{\text{max}}}{e^{\alpha \cdot w_k}}, \tag{11}$$

where $n_k$ is the number of source samples in class $k$, $n_{\text{max}}$ is the size of the largest class, $w_k$ is the adaptive weight of class $k$, and $\alpha$ is a temperature parameter. For each sample in class $k$, a uniform random variable $r \sim \mathcal{U}(0,1)$ is generated. The sample is selected for interpolation if $r > P_k$,

A smaller $P_k$ indicates that class $k$ is underrepresented or harder to align, thus increasing its chance of triggering interpolation. The rational behind this formulation ensures: (1) Minority classes with smaller $n_k$ naturally yield lower $P_k$ values, thereby enhancing their sampling priority; (2) Classes with higher difficulty (characterized by larger $w_k$) are also afforded higher priority via exponential scaling. As a result, This mechanism effectively addresses both inter-class imbalance and intra-class variability.

#### 3.4.2 ADAPTIVE INTERPOLATION

For each source sample $\boldsymbol{x}_s$ that triggered generation with label $y_s = k$, we sample a target pseudo-labeled sample $\boldsymbol{x}_t$ with $\hat{y}_t = k$ and generate an interpolated point:

$$\boldsymbol{x}_p = (1 - \lambda)\boldsymbol{x}_s + \lambda \boldsymbol{x}_t, \quad \lambda \sim \text{Beta}(a, b), \tag{12}$$

where $\lambda$ is drawn from a symmetric Beta distribution defined by two hyperparameters $a$ and $b$ to encourage interpolation near the midpoint while still allowing variance. This sampling strategy implicitly expands the feature support for minority classes and bridges fragmented cross-domain regions.

#### 3.4.3 ADVERSARIAL LEARNING-BASED EXPLORATION

To enhance the informativeness of generated samples, we formulate a bi-level adversarial optimization to search for the optimal $\lambda^*$ to explore the hard region. The outer optimization minimizes classification loss over generated samples:

$$\mathcal{L}_{\text{BPIG}}(\theta_f, \theta_g) = -\log \sigma_k(f(g(\boldsymbol{x}_p))), \tag{13}$$

where $k$ is the class index and $\sigma_k(\cdot)$ denotes the predicted probability for class $k$.

Simultaneously, the inner optimization maximizes the same loss by perturbing the interpolation coefficient $\lambda$:

$$\lambda^* = \arg \max_{\lambda \sim \text{Beta}(a,b)} \left[ -\log \sigma_k(f(g((1-\lambda)\boldsymbol{x}_s + \lambda \boldsymbol{x}_t))) \right], \tag{14}$$

encouraging the generation of harder samples near the decision boundary. This iteration process of adversarial interplay ensures that the interpolated samples are both realistic and challenging.

### 3.5 Total Objective Function

The overall training objective integrates multiple loss components to balance classification accuracy, domain alignment, and robustness to class imbalance. It consists of three key parts:

Supervised loss on labeled source samples to ensure basic classification capability:

$$\mathcal{L}_{\text{cls}}(\theta_f, \theta_g) = -\frac{1}{N_s} \sum_{i=1}^{N_s} \sum_{c=1}^{C} y_i^s(c) \log(f(g(\boldsymbol{x}_i^s))(c)) \tag{15}$$

where $y_i^s(c)$ is the one-hot label of source sample $i$ (1 if class $c$, else 0), and $f(g(\boldsymbol{x}_i^s))(c)$ is the classifier's predicted probability for class $c$.

To reduce source-target distribution gaps, we minimize the domain discrimination loss:

$$\mathcal{L}_{\text{domain}}(\{\theta_{D_i}\}, \theta_{D_{\text{global}}}) = \sum_{k=1}^{K} \beta_k \mathcal{L}_{\text{local},k} + \mathcal{L}_{\text{global}} \tag{16}$$

To enhance robustness via interpolated samples:

$$\mathcal{L}_{\text{BPIG}}(\theta_f, \theta_g) = \mathcal{L}_{\text{CE}}(\hat{x}_p, y; \theta_g, \theta_f) \tag{17}$$

where $\theta_g$ represents the parameters of the feature extractor $g$, and $\theta_f$ denotes the parameters of the classifier $f$.

The total objective function is a weighted combination of these losses:

$$\min_{(\theta_f, \theta_g)} \max_{(\{\theta_{D_i}\}, \theta_{D_{\text{global}}})} = \lambda_1 \mathcal{L}_{\text{cls}} + \lambda_2 \mathcal{L}_{\text{domain}} + \lambda_3 \mathcal{L}_{\text{BPIG}}. \tag{18}$$

## 4 Experiments

### 4.1 Datasets and Baselines

The experiment constructs a class-imbalanced cross-domain adaptation benchmark using two vision datasets: OfficeHome (Venkateswara et al., 2017) and Office31 (Saenko et al., 2010). OfficeHome contains 65 object categories and 4 domains: RealWorld , Clipart, Product, and Art. This experiment focuses on RealWorld, Product, Clipart, with 6 cross-domain tasks: RealWorld→Product (Rw→Pr), RealWorld→Clipart (Rw→Cl), Clipart→Product (Cl→Pr), Clipart→RealWorld (Cl→Rw), Product→RealWorld (Pr→Rw), Product→Clipart (Pr→Cl). Office31(Saenko et al., 2010) is a classic object recognition benchmark with 31 categories and 3 domains: Amazon, Webcam, DSLR. Focusing on all 3 domains, 6 tasks are designed: Amazon→Webcam (A→W), Amazon→DSLR (A→D), Webcam→Amazon (W→A), Webcam→DSLR (W→D), DSLR→Amazon (D→A), DSLR→Webcam (D→W).

To simulate the combined challenge of feature and label distribution shifts, we adopt the Reversely-unbalanced Source (RS) + Unbalanced Target (UT) protocol as shown in Fig. 2. Specifically, class imbalance is introduced via Pareto sampling in both domains, with reversed label distributions.

We selected the following Unsupervised Domain Adaptation (UDA) methods as comparative baselines: BSP(Chen et al., 2019), DANN(Ganin et al., 2016), F-DANN(Wu et al., 2019a), MCD(Saito et al., 2018), COAL(Tan et al., 2020), CDAN-E(Long et al., 2018), Sentry(Prabhu et al., 2021), MDD(Zhang et al., 2019), MDD+implicit(Jiang et al., 2020) and MDD+PAT(Shi et al., 2022a).

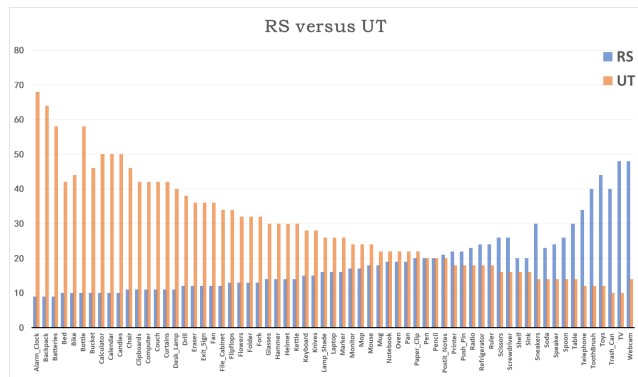

Figure 2: Reversely-unbalanced Source (RS) v.s. Unbalanced Target (UT) protocol.

## 4.2 RESULTS ANALYSIS

As shown in Table 1, ALFADA consistently outperforms all baselines under the RS→UT protocol on OfficeHome. It achieves an average accuracy of 66.92%, improving upon the strongest baseline (MDD+PAT) by 0.97%. Notably, in Rw→Pr and Pr→Rw, ALFADA yields gains of 1.08% and 1.12%, respectively. Even under large domain gaps (e.g., Rw→Cl), it surpasses prior methods, demonstrating superior adaptability to both domain and label shift. These improvements stem from the framework's integrated mechanisms: adaptive sampling prioritizes informative instances, the local discriminative forest captures regional domain gaps, and boosted interpolation enriches sparse feature manifolds—collectively enabling robust cross-domain adaptation under complex distributional skew.

As shown in Table 2, ALFADA also achieves state-of-the-art results on the Office-31 dataset under the RS→UT protocol, with an average accuracy of 86.63%, surpassing the strongest baseline (MDD+PAT) by 1.34%. The improvements are particularly notable on challenging transfer tasks such as A→W (+1.45%) and A→D (+1.46%), demonstrating the framework's strength in handling both domain and label shift. Even on relatively easier transfers (e.g., W→D and D→W), ALFADA continues to deliver gains, indicating that its adaptive sampling and local discriminative mechanisms provide consistent benefits across varying levels of adaptation difficulty. These results further confirm the generalizability and robustness of ALFADA beyond a single dataset.

To validate the rationality of leveraging per-class MERs to guide the whole training adaptively, we use Figure 3 to visualize the correlation between MER and final classification accuracy on target domains. As MER decreases, corresponding target classification accuracy consistently improves. This trend confirms matching error rate effectively quantifies cross-domain alignment quality. By guiding sampling and alignment during the whole training based on matching error rate, ALFADA rationally focuses on minority instances and bridge domain shift and classification robustness.

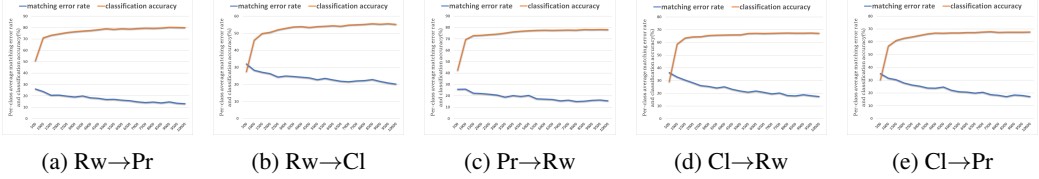

| (a) Rw→Pr | (b) Rw→Cl | (c) Pr→Rw | (d) Cl→Rw | (e) Cl→Pr |

Figure 3: Per-class average matching error rates and the corresponding classification accuracies.

## 4.3 ABLATION STUDY

To verify the effectiveness of each component in ALFADA, we conduct ablation experiments. We use the DANN as our base model and evaluate Local Discriminative Forest (LDF) and Boosted Pairwise Interpolation Generator (BPIG) by comparing the full model against variants with each component removed. The setup is consistent with our main experiments, using the same dataset

Table 1: Per-class average accuracy (%) on Office-Home dataset with RS→UT label shift.

| Method | Rw→Pr | Rw→Cl | Pr→Cl | Pr→Rw | Cl→Rw | Cl→Pr | AVG |
|---|---|---|---|---|---|---|---|
| BSP | 72.74 | 23.88 | 20.09 | 66.26 | 32.79 | 30.41 | 41.03 |
| DANN | 71.89 | 46.05 | 35.78 | 68.06 | 58.69 | 57.83 | 56.38 |
| F-DANN | 58.56 | 40.57 | 37.29 | 67.32 | 55.84 | 53.67 | 44.24 |
| MCD | 66.34 | 32.19 | 29.74 | 62.59 | 41.47 | 38.75 | 45.18 |
| COAL | 73.68 | 42.64 | 38.16 | 73.29 | 59.36 | 57.27 | 57.40 |
| CDAN-E | 70.85 | 45.91 | 36.43 | 69.83 | 53.96 | 54.82 | 59.07 |
| Sentry | 76.14 | 56.77 | 48.69 | 73.62 | 65.96 | 64.33 | 64.25 |
| MDD | 71.18 | 44.75 | 42.26 | 69.35 | 52.12 | 52.68 | 55.42 |
| MDD+implicit | 76.10 | 50.08 | 45.73 | 74.31 | 61.11 | 63.20 | 61.76 |
| MDD+PAT | 79.29 | 54.63 | 50.17 | 77.25 | 67.21 | 67.15 | 65.95 |
| **ALFADA(Ours)** | **80.37** | **55.71** | **51.41** | **78.37** | **67.69** | **67.98** | **66.92** |

Table 2: Per-class average accuracy (%) on Office-31 with RS→UT label shift.

| Method | A→W | D→W | W→D | A→D | D→A | W→A | AVG |
|---|---|---|---|---|---|---|---|
| BSP | 71.77 | 90.86 | 93.06 | 72.25 | 59.03 | 58.34 | 74.21 |
| F-DANN | 69.83 | 93.56 | 93.95 | 76.45 | 58.57 | 58.11 | 75.07 |
| COAL | 81.18 | 91.12 | 95.46 | 81.67 | 66.08 | 66.60 | 80.35 |
| CDAN-E | 76.25 | 95.78 | 94.85 | 79.92 | 64.04 | 58.69 | 78.25 |
| Sentry | 82.01 | 90.86 | 93.73 | 83.55 | 62.79 | 63.88 | 79.47 |
| MDD | 83.99 | 96.69 | 96.71 | 83.94 | 67.23 | 61.36 | 81.65 |
| MDD+implicit | 85.79 | 96.20 | 97.40 | 84.25 | 68.11 | 66.63 | 83.06 |
| MDD+PAT | 89.61 | 96.08 | 97.08 | 86.66 | 71.93 | 70.40 | 85.29 |
| **ALFADA(Ours)** | **91.06** | **97.42** | **97.70** | **88.12** | **73.49** | **71.98** | **86.63** |

splits, per-class accuracy metric, and hyperparameters. As table 3 illustrated, removing the Local Discriminative Forest will cause performance drops across all tasks, with the most significant declines in Rw→Cl (will drop by 0.99%) and Cl→Pr (will drop by 0.97%). This indicates LDF's role in capturing fine-grained regional discrepancies, critical for aligning sparse minority features overlooked by global alignment. Removing BPIG will lead to an average drop of 4.76% due to the reason that BPIG enriches sparse regions via source-target interpolations, enhancing decision boundary robustness for underrepresented classes. These results demonstrate Local Discriminative Forest enables fine-grained alignment, and BPIG enriches sparse regions, collectively addressing class imbalance and domain shift.

Table 3: Per-class average accuracy(%) of DANN, ALFADA w/o BPIG, ALFADA w/o Local Discriminative Forest(LDF), and ALFADA (full) on imbalanced Office-Home dataset.

| Method | Rw→Pr | Rw→Cl | Pr→Cl | Pr→Rw | Cl→Rw | Cl→Pr | AVG |
|---|---|---|---|---|---|---|---|
| DANN | 71.89 | 46.05 | 35.78 | 68.06 | 58.69 | 57.83 | 56.38 |
| ALFADA w/o BPIG | 76.8 | 51.03 | 41.39 | 72.54 | 66.21 | 64.97 | 62.16 |
| ALFADA w/o LDF | 78.89 | 54.72 | 48.12 | 77.95 | 66.97 | 67.01 | 65.61 |
| **ALFADA (full)** | **80.37** | **55.71** | **51.41** | **78.37** | **67.69** | **67.98** | **66.92** |

## 5 CONCLUSION

In this paper, we propose ALFADA, a novel adversarial domain adaptation framework tailored for scenarios with severe class imbalance and label distribution shift. By leveraging adaptive instance weighting based on matching error rates, constructing a local discriminative forest for region-specific alignment, and generating interpolated samples via a Boosted Pairwise Interpolation Generator (BPIG), ALFADA effectively addresses both global and local domain discrepancies. Extensive experiments on the Office-Home and Office-31 benchmarks demonstrate that ALFADA consistently outperforms state-of-the-art methods across all transfer tasks.

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
