# OpenReview forum: "Adaboost-Based Local-Forest Adversarial Learning for Imbalanced Domain Adaptation"
_ICLR.cc/2026/Conference — ICLR 2026 Conference Desk Rejected Submission_

### Official Review · Reviewer_4rAT · 2025-10-28

**Soundness:** 2
**Presentation:** 1
**Contribution:** 1
**Rating:** 2
**Confidence:** 3

**Summary:**

This paper addresses the unsupervised domain adaptation (UDA) problem, where labeled source-domain data and unlabeled target-domain data are available. The authors propose a new framework, named ALFADA, designed to handle class imbalance in UDA. The approach combines three main components:
(1) a weighted sampling scheme inspired by AdaBoost,
(2) a set of local discriminators constructed via clustering in a latent feature space, and
(3) a data synthesis module that interpolates between high-weighted source instances and confident target samples of the same predicted class, where the interpolation coefficients are optimized adversarially.

**Strengths:**

The paper tackles a meaningful and practically important challenge — class imbalance in domain adaptation. The proposed framework attempts to integrate ideas from boosting, adversarial training, and data augmentation in an interesting way.

**Weaknesses:**

1. **Poor writing and organization.**  The paper is difficult to follow due to unclear exposition and inconsistent terminology. For instance, the proposed method is first referred to as _ALFAL_ in the abstract but later as _ALFADA_, suggesting a lack of proofreading and editorial care. Figure 1 occupies significant space yet fails to illustrate the proposed framework effectively, nor is it adequately explained in the text. Repetitions (e.g., Lines 186 and 197) and abrupt term introductions further hinder readability.
2. **Unclear motivation and intuition.**  The rationale behind the design of key components—e.g., Equations (1)–(3) and (14)—is not well justified. The paper presents a complex combination of techniques (weighted sampling, clustering, adversarial interpolation) but does not clearly explain why each component is necessary or how they interact to address class imbalance.
3. **Inconsistent and undefined terminology.**  Many ad hoc terms appear without proper definition or explanation.
	- “Matching rate” (Line 173) is introduced before being defined, and later replaced by “matching error rate,” creating confusion.
	- “Domain discriminator” (Line 234), “local domain mismatch,” and “enhance feature continuity” (Line 285) are mentioned without prior explanation or formal definition.
	- The meaning of “global distribution shift” versus “local feature region alignment” (Line 279) remains vague.
	These inconsistencies make it difficult to interpret the proposed approach.


4. **Notation issues and unclear formulation.**  In Equation (14), the argmax operator appears to be applied over a random variable or distribution, which is mathematically unclear. The purpose and implications of this formulation should be clarified.

The paper lacks coherence and polish. The logical flow of sections (method, objective, and analysis) is weak. As a result, it is challenging to extract a precise understanding of the proposed contributions.

**Questions:**

- Could you elaborate on the intuitions behind the design of Equations (1)–(3)?
- What is meant by _“global distribution level shift”_? Why is it necessary to decompose domain alignment into local and global levels, and how does this help address class imbalance?
- For Objective (14), what is the rationale for perturbing the interpolation coefficients rather than the instances themselves, as done in conventional adversarial training?

---

### Official Review · Reviewer_gYgh · 2025-10-29

**Soundness:** 2
**Presentation:** 2
**Contribution:** 2
**Rating:** 4
**Confidence:** 4

**Summary:**

This paper proposes a method called Adaboost-based Local-Forest Adversarial Domain Adaptation (ALFADA) for addressing the Class-Imbalanced Domain Adaptation (CDA) problem. To tackle issues such as class imbalance, pseudo-label noise, and fine-grained domain misalignment, the authors introduce three modules: adaptive matching error rate sampling, Local Discriminative Forest (LDF), and Boosted Pairwise Interpolation Generator (BPIG). Experimental results demonstrate the effectiveness of the proposed method.

**Strengths:**

The paper clearly explains the rationale and purpose of each step in the method section, which facilitates the reviewer’s understanding and makes the overall article clear and easy to follow.

**Weaknesses:**

1. The overall novelty of the paper is limited. In my view, the authors' main innovation lies in designing sample or category weights that correspond to the samples and loss function. However, other aspects, such as adversarial learning and sample interpolation, are well-established techniques.
2. The abstract is poorly written. It does not clearly state the core problem or how the authors address it.
3. All issues mentioned in the "questions" section.

**Questions:**

1. After reading the entire paper, I noticed that pseudo-label supervised learning for target domain samples is not used. Compared to adversarial training, this is often a more effective learning method. Why didn’t the authors consider using it? The target domain samples above the threshold are considered confident, and these samples could be used for learning.
2. Target domain samples with low confidence seem to be excluded from training entirely, which discards a significant amount of useful information. Proper utilization could bring improvements. I believe this is a design flaw.
3. Regarding Section 3.2, Equation 1 seems redundant. Pseudo-labels are outputs from the previous epoch, and the samples are all high-confidence, meaning that the predicted labels will always match the pseudo-labels, resulting in an error rate of zero. Even if the error rate isn’t zero, Equation 2 could still yield a positive or negative value, depending on whether the error rate exceeds 0.5. Therefore, Equation 3 might actually reduce the weight. Moreover, as the error rate increases, αc becomes smaller, which decreases the weight wi for misclassified samples. This contradicts the authors’ explanation.
4. The clustering section is puzzling. If the source and target domains are not aligned, how can they be accurately clustered into K categories? This would introduce a lot of noise, leading to failure in subsequent adversarial alignment.
5. How is the validation set selected in Section 3.3.2?
6. Where does wk come from in Equation 11?
7. In Equation 12, λ is supposed to be a sample, but in Equation 14, it appears to be obtained through adversarial learning. How does this transition happen?
8. In Equation 17, hat(xp) is written. It should correspond to Equation 12.

---

### Official Review · Reviewer_HRSN · 2025-11-01

**Soundness:** 2
**Presentation:** 3
**Contribution:** 2
**Rating:** 2
**Confidence:** 3

**Summary:**

The paper addresses a critical issue in Unsupervised Domain Adaptation (UDA) known as Class-Imbalanced Domain Adaptation (CDA). The class imbalance exacerbates the domain distribution shift, causing adaptation models to become biased towards majority classes and perform poorly on minority ones. The authors propose a unified framework named ALFADA. The core idea is to leverage the property of the AdaBoost algorithm by implementing an adaptive mechanism that identifies and concentrates on hard-to-align feature regions and under-represented minority-class samples.

**Strengths:**

1. A key highlight is the integration of the AdaBoost principle (dynamic reweighting to focus on difficult instances) across the entire framework, from sample selection and local alignment to data generation.
2. The integration of Local Discriminative Forest (LDF) is a good design. By partitioning the feature space via weighted clustering and assigning specialized local discriminators, it captures fine-grained discrepancies that a global discriminator would miss.
3. Pairwise Interpolation Generator introduced linear combination of data generation and the adversarial search for the optimal interpolation coefficient $\lambda$ enhances classifier robustness.

**Weaknesses:**

1. The proposed ALFADA framework is considerably complex and introduces a large number of hyperparameters, e.g., the confidence threshold, number of clusters, clustering frequency, Beta distribution parameters and so on. This work lacks a detailed discussion on the selection criteria or a sensitivity analysis for these parameters, which could pose challenges for reproducibility.
2. Combined with the above weakness, the complex framework with large amount of hyperparameters and the min-max bi-level optimization would lead to the substantially higher computational costs. The absence of an analysis on training efficiency makes it difficult to assess the method's practicality on large-scale datasets.
3. Multiple stages of the framework depend on the quality of pseudo-labels generated for the target domain. Although a confidence threshold is used for filtering, performance may suffer from error accumulation in the early stages of training or in tasks with very large domain gaps. The ablation study on the quality of pseudo-labels should be discussed.
4. The framework contains the inner-loop adversarial optimization required to find optimal $\lambda^*$ for each sample generated by BPIG. Is this process guaranteed to converge, or improves the oscillation during training?
5. Regardless of the method design, the reviewer would like to question the reproducibility of this work, since there is no discussion about the experimental settings and details. The authors are expected to discuss the details during the rebuttal time.

**Questions:**

Please see above weaknesses.

---

### Official Review · Reviewer_uhpw · 2025-11-03

**Soundness:** 2
**Presentation:** 2
**Contribution:** 2
**Rating:** 2
**Confidence:** 4

**Summary:**

The paper proposes an AdaBoost-based Local-Forest Adversarial Learning framework to address the class imbalance problem in unsupervised domain adaptation, with a particular focus on hard-to-align samples.

**Strengths:**

1. Exploring the imbalance domain adaptation problem with a focus on hard-to-align samples is an interesting direction.
2. The method is well illustrated.

**Weaknesses:**

1. The paper does not provide a clear illustration or definition of what constitutes a hard-to-align sample
.
2. The novelty of the proposed method appears limited. The proposed local/global discriminator and boosted pairwise interpolation generator are very similar to prior works, and the authors fail to adequately justify the differences or contributions compared to existing methods.

3. The performance improvement of the proposed method is marginal compared to baselines.

4. No ablation study is conducted.

**Questions:**

see weakness

---

### Note · Program_Chairs · 2026-01-17
**Submission Desk Rejected by Program Chairs**

The following references in this submission do not refer to real documents and/or have major errors in bibliographic information:

 Bin Huang, Lin Ding, Jiayi Zhou, et al. Gradient-based dynamic reweighting for class-imbalanced semantic segmentation. arXiv preprint arXiv:2407.01327, 2023.